# Primary Soft Tissue Sarcoma of the Heart: An Emerging Chapter in Cardio-Oncology

**DOI:** 10.3390/biomedicines9070774

**Published:** 2021-07-03

**Authors:** Pietro Scicchitano, Maria Chiara Sergi, Matteo Cameli, Marcelo H. Miglioranza, Marco Matteo Ciccone, Marica Gentile, Camillo Porta, Marco Tucci

**Affiliations:** 1Cardiology Department, Hospital “F. Perinei”, 70022 Altamura, Italy; 2Department of Biomedical Sciences and Human Oncology, University of Bari Aldo Moro, 70124 Bari, Italy; sergimariachiara@gmail.com (M.C.S.); marica.gentile@libero.it (M.G.); camillo.porta@uniba.it (C.P.); marco.tucci@uniba.it (M.T.); 3Department of Medical Biotechnologies, Division of Cardiology, University of Siena, 53100 Siena, Italy; matteo.cameli@yahoo.com; 4Cardiology Institute of Rio Grande do Sul, Universidade Federal de Ciências da Saúde de Porto Alegre (UFCSPA), Porto Alegre 90050-170, Brazil; marcelohaertel@gmail.com; 5Cardiology Section, Department of Emergency and Organ Transplantation, University of Bari Aldo Moro, 70124 Bari, Italy; marcomatteo.ciccone@uniba.it

**Keywords:** cardiac sarcoma, diagnosis, molecular diagnosis, treatment, cardio-oncology

## Abstract

Primary malignant cardiac tumors are rare, with a prevalence of about 0.01% among all cancer histotypes. At least 60% of them are primary soft tissue sarcomas of the heart (pSTS-h) that represent almost 1% of all STSs. The cardiac site of origin is the best way to classify pSTS-h as it is directly linked to the surgical approach for cancer removal. Indeed, histological differentiation should integrate the classification to provide insights into prognosis and survival expectancy of the patients. The prognosis of pSTS-h is severe and mostly influenced by the primary localization of the tumor, the difficulty in achieving complete surgical and pharmacological eradication, and the aggressive biological features of malignant cells. This review aims to provide a detailed literature overview of the most relevant issues on primary soft tissue sarcoma of the heart and highlight potential diagnostic and therapeutic future perspectives.

## 1. Introduction

Primary soft tissue sarcomas of the heart (pSTS-h) are rare and represent an interesting chapter in cardio-oncology [1]. Autoptic data revealed a prevalence of pSTS-h ranging from 0.001% to 0.03% [2,3,4]. The pSTS-h are difficult to be diagnosed, and retrospective data exploring 7,384,580 cases of cancer included in the Surveillance, Epidemiology and End Results (SEER)-18 database from the United States National Cancer Institute from 1973 to 2015 revealed about a 0.01% prevalence [5]. Other epidemiologic investigations [1], however, outlined a 10.8% prevalence of pSTS-h among 8800 patients with primary cardiac tumors. Despite their rarity, primary heart cancers show high mortality with almost 15% of deaths [1], and STS is the most common histotype [5], which includes at least 1% of all sarcomas [6]. The prognosis is severe and mostly influenced by the localization of the tumor, its biological aggressiveness, and the difficulty in obtaining surgical radicality, whereas chemotherapy exerts a modest role [7]. The one-year overall survival of pSTS-h is 25% [8], whereas angiosarcoma, leiomyosarcoma, and poorly differentiated sarcoma are the most aggressive variants, showing worse prognosis and risk of death in more than 70% of patients [8]. In this context, radical surgery, chemotherapy, and radiotherapy may improve both progression-free survival (PFS) and overall survival (OS) [8], although the majority of data have been collected from single-center experience or retrospective analyses.

Herein, recent issues on pSTS-h and either diagnostic or therapeutic future perspectives are described.

## 2. Classification, Pathological Features, and Molecular Features

The classification of pSTS-h is still under debate. The rarity of the disease and the lack of tailored clinical studies render the evaluation of rare histotypes less comprehensive. The pSTS-h classification is based on the cardiac site of origin, such as those arising from the right, left chamber, pericardium, or cava vein and aorta [9,10,11,12,13]. The early detection of the sarcoma mass is critical for planning the best surgical approach, and thus the proposal to combine surgery and histologic evaluation (Figure 1), accordingly with the 5th Edition of the World Health Organization (WHO) Classification, is a relevant strategy for planning the best treatment of pSTS-h.

Briefly, the STSs from the great vessels (aorta/great veins) are differentiated into (i) mural and (ii) intimal in relation to their development from the mid-layer/adventitia of the vascular wall or the intima, respectively [9,12]. Moreover, lesions arising from the intima usually invade the lumen of the vessels, while mural pSTSs tend to outer extension. Furthermore, intimal STSs often show poorly differentiated histology that negatively impacts outcomes in the majority of patients [9,12]. On the basis of the grade of differentiation (Figure 1), one can subdivide the pSTSs of the right and left chambers of the heart into low, intermediate, and high grades.

The difficulties in diagnosis and the aggressiveness of these mass induce the outer cells to widely disseminate to different organs. Lungs are the main site for metastases from pSTS-h, followed by lymph nodes, bones, liver, and the central nervous system [14]. This is in agreement with other forms of sarcomas arising from tissues other than the heart: the main final destination of the metastasis still remains the lungs, but all the organs and tissues may become the target of disseminated cells of sarcoma.

### 2.1. Angiosarcoma

Angiosarcoma (A-SRC) is the most frequent histologic variant characterized by an incidence of 7.3–8.5% among all tumors [3,15,16]. Epidemiologic data from the Japanese Circulation Society and the Japanese Association for Thoracic Surgery reported incidences of 8.2% and 9.5% during the last decade [15,17]. A-SRCs preferentially develop from the endothelial layer of the cardiac chambers [17,18,19], and Leduc et al. [20] outlined that they are most likely to arise from the right atrium followed by epicardium and right ventricle. They may grow up to 10 cm in diameter, often infiltrating the borders, thus resulting in difficulty in being distinguished from the cardiac wall [20]. Accordingly with morphology and invasiveness, A-SRCs may develop from the cardiac chambers as large, bleeding, or necrotic lesions showing a high propensity to infiltrate the pericardium and surrounding structures [18,21]. The cardiac A-SRCs can effectively invade the cardiac chambers, thus provoking obstruction and heart failure as well as clinical complications related to the compression of the great veins or invasion of the pericardium [4,22]. Malignant cells from A-SRCs usually show pleiomorphic nuclei and high mitotic count [23]. Spindle-cell is the most common histotype, although epithelioid cells [20] showing a focal distribution of cytokeratins are frequently found [20]. Indeed, vascular markers, such as ERG (erythroblast transformation-specific (ETS)-related gene), a member of the ETS family of transcription factors such as both the avian v-ets erythroblastosis virus E26 oncogene homolog and [24] the friend leukemia virus integration-1 (FLI-1), as well as CD31 frequently characterize A-SRC of the heart [20,23]. The molecular cytogenetic landscape does not foster a definite and specific diagnosis [20]. However, trisomy of chromosomes 4, 8, 11, 17, and 20 have been previously demonstrated in parallel to the amplification in *MYC*, *PIK3C2B/MDM4*, and *KIT* [20].

### 2.2. Undifferentiated High-Grade Pleomorphic Sarcoma

Undifferentiated high-grade pleomorphic cardiac sarcomas (UHGPC-SRCSs, also named as malignant fibrous histiocytomas) are the second pSTS-h (Table 1). 

Among the pSTS-h cases from the French experience, 36% were UHGPC-SRCSs [8]. They deeply invade the myocardium and surrounding structures, often generating large necrotic areas. It can appear as a sessile or pedunculated mass [19]. The cytology is characterized by the presence of different cells with spindle and epithelioid or giant features. Such cells often display a higher number of mitoses and nuclear polymorphism, while others undergo late apoptosis and necrosis. Moreover, UHGPC-SRCSs are formed by poorly differentiated mesenchymal or chondromatous phenotypes that confer a high degree of malignancy [7]. The heterogeneous features of these neoplasms make the definite diagnosis difficult [25]. Indeed, the classification of UHGPC-SRCSs requires immunohistochemistry, and vimentin expression is considered reliable for this type of pSTS-h [8], as well as α-smooth muscle actin and cytokeratins [8]. In addition, chromosomal defects of *AKT2* and *RUNX2*, mutation of *PDGFRB*, and a higher number of chromosomal aberrations have been described in these rare variants of SRCs [26].

### 2.3. Rhabdomyosarcoma

The rhabdomyosarcoma of the heart (RMS) mostly occurs in children and adolescents, showing the highest incidence in 15-year-olds [22]. In adulthood, the Armed Forces Institute of Pathology (AFIP) registry revealed an incidence ranging from 1.6% to 4.9%, although the identification of such neoplasms after surgical excision was 0.07%, the incidence ratio being higher in children < 16 years old (5.4%) [3,15,16]. Indeed, the Japanese experience outlined the lowest incidence, ranging from 0.45% to 0.8% [15,17]. As this sarcoma seems to originate from the degeneration of embryonic cells, its frequency in younger age [7,22] is, at least in part, explained. There is no specific cardiac location for RMS, although they may often arise from the myocardium of the ventricle—the left in particular—but rarely from the atria [7,27]. The mass may develop throughout the heart, therein involving the cardiac valves and/or the pericardium, where it may appear as a nodule [21,22]. Macroscopically, RMS arises from the myocardium as large, irregular, infiltrative masses, also characterized by necrotic areas [21], while the infiltration of the pericardium may lead to hemorrhagic infiltration [23]. Microscopically, RMS is usually formed by cells that resemble the rhabdomyoblasts with pleiomorphic nuclei frequently resulting in positive to α-smooth muscle actin, desmin, myogenin, and vimentin [7].

### 2.4. Leiomyosarcoma

Cardiac leiomyosarcoma (LMS) is rare, and the French experience reported a 12.9% prevalence [8]. Most LMSs are located in the left atrium [19]. Indeed, these sarcomas promptly disseminate to pulmonary veins or invade mitral leaflets [28]. This can account for the symptoms of patients, while the survival rate remains poor [29]. The histological pattern can reveal the presence of abnormal smooth muscle cells, which appear as spindle cells with blunt-ended or “cigar-shaped” nuclei [7,28]. Necrotic and pleomorphic areas can often be detected [7,28]. As the cardiac LMSs are neoplasms arising from the smooth muscle cells and still keep their differentiation status, histology can point out positivity to desmin, α-smooth muscle actin, and myogenin, whereas they rarely show positive and focal stain for cytokeratins [7,30].

### 2.5. Synovial Sarcoma

The incidence of cardiac synovial sarcomas is truly rare (Table 1). Most of them occur in individuals younger than 20–40 years old [31]. Cardiac synovial sarcoma usually develops in the atria and pericardium [7,28,32]. The right atrium is the most common site, while the pericardial localization requires differentiation from primary mesothelioma [7]. The invasiveness of the growing mass leads to the fast invasion of other cardiac chambers and great dimensions of the malignant neoplasm that can be 10–15 cm in diameter [32]. Microscopic evaluation often revealed both epithelial and spindle cells differently mixed within the malignant tissue [7,28,32]. The presence of these cells or the identification of spindle cells accounts for the biphasic or monophasic variant, respectively [28,32]. The cytology outlines the presence of cells with modest cytoplasm and nuclei bearing fine chromatin [32]. Mitoses can also be detected, while necrotic areas can predominate in more undifferentiated forms [32]. The identification of hemangiopericytoma-like vascular structures is a distinctive feature for cardiac synovial sarcomas compared to those arising from other sites [33,34]. The genetic background of cardiac synovial sarcomas pointed out the possibility of translocation of chromosome X to chromosome 18 (p21.2; q11.2), which promotes the fusion between *SYT* on chromosome 18 and *SSX1* or *SSX2* on chromosome X [7]. Indeed, cytokeratins and vimentin can be detected, while rarely malignant cells express epithelial membrane antigen (EMA) and/or α-smooth muscle actin [7].

### 2.6. Liposarcoma

Primary cardiac liposarcoma (LSRC) is a rare entity with an incidence of 0.19–0.5% among all cardiac neoplasms, according to the AIFP [3,15,16], while the Japanese registry reports a 0.9–1.1% incidence [15,17]. Right cardiac chambers are the preferred site of origin, whereas the diffusion to pericardium often leads to the development of nodules or irregular masses, or malignant effusion [25]. The macroscopic expression of this cardiac neoplasm is multilobulated with necrotic or hemorrhagic areas [25]. Three types of cardiac LSRC can be distinguished according to the degree of cellular differentiation: myxoid, dedifferentiated, and pleiomorphic [35]. Immunohistochemistry revealed positivity to the S100 protein and vimentin [7].

### 2.7. Fibrosarcoma and Myxoid Fibrosarcoma

The incidence of fibrosarcoma (FSRC) and myxoid fibrosarcoma (M-FSRC) is about 2.3–3.2% (AFIP registry) [3,15,16], although Japanese registries report an overall incidence between 0 and 0.3% among all forms of cardiac neoplasms [15,17]. Most of them arise from the left atrium, although other sites may be rarely involved [25]. Fibrosarcomas are characterized by spindled cells that show fascicular patterns in a collagenous stroma with numerous large hyalinized collagen rosettes. When myxoid features prevail, the term myxoid can be added to define the M-FSRC [28]. Furthermore, patterns of M-FSRC are often characterized by high activation of features of neo-angiogenesis [28]. The immunohistochemistry outlines the possible expression of vimentin and, sometimes, α-smooth muscle actin [28,36]. The translocation t(7; 16) (q33; p11) able to promote the fusion between *FUS* and *CREB3L2* genes seemed to occur more often in such cardiac neoplasms [36].

### 2.8. Osteosarcoma and Chondrosarcoma

Primary cardiac osteosarcomas and chondrosarcoma are rare malignant entities (Table 1) that mostly occur in young/middle-aged individuals (20–60-year-old) [37]. The literature provides poor data about these two pathological, infiltrative, and aggressive entities, except for some case reports [37,38,39,40]. The left atrium is the preferred site of origin, thus explaining the symptoms generated by this neoplasm [19,28]. Clearly, the masses can reach great dimensions and diameters ranging from 2.5 to 13 cm have been previously reported. They rise as pedunculated mass that protrudes into the left atrium, invade the mitral leaflets, and penetrate the walls of the atrium. The masses usually manifest as gelatinous or mucoid, while osteosarcomas can demonstrate calcified areas. The surface of these neoplasms may become rough or smooth, while areas with necrosis or hemorrhagic spots can also be observed. Osteoblastic, chondroblastic, or fibroblastic types of cells are detectable at the histological analysis of the specimens. The cancer is composed of spindle cells with areas of osteoid, sometimes with undifferentiated elements or giant cells [19,37,38].

## 3. The Role of Echocardiography in pSTS-h

The evaluation of patients with cardiac tumors—and pSTS-h in particular—is challenging. As signs and symptoms related to the growing masses may not be specific, the first step for diagnosis should be based on imaging techniques. A multimodality approach should be considered for a detailed, graphical reconstruction of the abnormal mass and the evaluation of the relationships between cancer and the neighbor tissues [7,12]. A stepwise approach is the best way to correctly identify the outer mass or, at least, to guide interventions.

Transthoracic bi-dimensional echocardiography (TTE) is the baseline technique for diagnosis. TTE is a useful, fast, costless, non-invasive tool able to address the evaluation of pSTS-h and other forms of cardiac cancer at the bedside [41]. The TTE allows for the localization of the mass into the cardiac chambers and/or great arterial/venous vessels; the identification of the involved cardiac structures—i.e., valves, papillary muscles/tendons chordae, cardiac walls; the measurement of their dimension; and the description of their mobility [25]. The reported sensitivity value of TTE ranges from 75% to 93.3% when applied to the identification of cardiac tumors [42,43]. Indeed, it is difficult to distinguish between benign and malignant masses. Furthermore, artifacts and poor acoustic windows can negatively influence diagnosis, while TTE is not able to distinguish the tissue histology of the outer mass [41,44]. The main echocardiographic characteristics of the pSTS-h can be briefly summarized as follows: the localization is the right side of the heart, the masses often involving the pericardium where they may promote pericardial effusion, sometimes degenerating into compression and cardiac tamponade [44]. Serial TTE evaluations may point out the fast growth of the pSTS-h, in contrast with benign forms such as myxomas. Furthermore, pSTS-h usually rises from the cardiac walls as lobulated masses with no peduncle, which allows for a possible differential diagnosis to benign masses such as myxomas. Most pSTS-h may show hypoechoic areas that are the ultrasonographic expression of necrotic/hemorrhagic zones [43,44].

Transoesophageal echocardiography (TOE) can improve the diagnostic power of ultrasonography. The higher spatial and temporal resolution of TOE, the adherence of the beam to the cardiac chambers, and the avoidance of transthoracic impedance artifacts are responsible for the higher sensitivity of the TOE as compared to TTE (96% vs. 93.3%, respectively) [42]. TOE allows for the intraoperative guidance of biopsies and a better definition of the images. Nevertheless, TOE is also an invasive procedure, thus including risk for patients’ health, and requires local expertise both in technical performance and identification of the cardiac tumors.

Contrast echocardiography can be an interesting option for implementing the correct identification of the masses. The infusion of contrast agent (microscopic particles comprised of gas-filled aqueous shells, for example) and the optimization of the images may display the vasculature—and the neo-vasculature—of cancer [45,46,47]. Malignant cardiac cancers can show abnormal vascularization as compared to benign masses such as myxomas, which are often poorly vascularized. Indeed, neo-vasculature is not a specific characteristic of pSTS-h—most of them have embryonic vessels, often incompletely generated [43]. The lack of contrast enhancement is the consequence of the presence of vessels in their early stages of development, thus avoiding correct discrimination between benign and malignant masses [43]. The need for adoption of contrast echocardiography lies in the possibility to better identify the characteristics of cancer in case of positive enhancement—the definition of the border and the inner vascularization might improve the diagnostics and visualization of masses as well as their tight relationship with cardiac walls [43,45]. Contrast echocardiography may also help physicians in discriminating intracardiac thrombi from abnormal growing masses [46,47]. The differential diagnosis has clinical and therapeutic implications: first, the decision about surgical approaches, and second, the need for anticoagulation, which is indicated in the presence of thrombi, but detrimental in the absence of coagula and to treat cancer.

Three-dimensional echocardiography (3D-echo) is the best approach for evaluating cardiac tumors and pSTS-h in particular. 3D-echo is effectively able to provide the best views of the cardiac chambers and abnormal masses by integrating the most innovative echocardiographic technique [48]. The 3D approach is reasonably able to correctly identify the dimension and function of cardiac chambers as well as cardiac magnetic resonance imaging (Figure 2) (cMRI) [48].

The application of 3D-echo to cardiac masses allows for the exact assessment of the dimensions (Figure 3 and Figure 4).

Asch et al. [49] observed the accurate evaluation of the mass volume due to the 3D-echo to represent the effective shape of the tumor within the three dimensions of the space. As TTE and TOE underestimate cardiac mass size by about 25% and 20%, respectively, as compared to 3D-echo, this latter technique is the best way to evaluate pSTS-h [49]. The analysis of the 3D images allows for the correct identification of the cleavage plane of the mass, thus promoting the delineation and surgical management of cancer, above all in case of malignant pSTS-h [50]. Furthermore, 3D-echo may display the masses within the right ventricle better than common 2D techniques [51].

Therefore, although a multimodality imaging approach is the best method for the complete evaluation of a patient suffering from pSTS-h, echocardiography remains the best, first-line method. Advance in technique can describe with details the outer masses since bedside, the technique being well tolerated by the patients, costless, almost reproducible, with no contraindication (contrast allergy, acute kidney failure due to contrasts, etc.), and widely available in peripheral as well as hub-hospitals.

## 4. cMRI in pSTS-h

Cardiac MRI represents the most important tool for the diagnosis of primary sarcomas in general, and pSTS-h in particular. It represents the most powerful non-invasive technique able to explore morphology and function of the heart with higher precision [52,53]. On parallel, cMRI demonstrated high temporal resolution while the use of late-gadolinium enhancement promoted the tissue characterization of the cardiac tissue (Figure 2) [52,53].

cMRI promotes the evaluation of the right cardiac chambers. Right cardiac chambers are the neglected chambers in the general assessment of the heart as right atrium and ventricle are difficult to correctly visualize with echocardiography [54]. cMRI overcomes the limitations of echocardiography and provides sequential analysis of right chamber morphology, tissue characterization, function, and reciprocal relationship with neighbor organs [54].

Mousavi et al. demonstrated that cMRI successfully distinguished benign from malignant cardiac cancer masses, as blinded readers were able to identify the correct mass in 89% and 94% of cases, respectively, thus demonstrating the great impact of the technique in stratification of the patients and their lesions [55].

cMRI promotes comparable results with positron emission tomography (PET). PET is adopted for the identification of the metabolism of the masses and their replicative activity. cMRI and the application of late-gadolinium enhancement are able to identify the avascular zones of the outer mass, which correlated with data from PET analysis [56]. The identification of viable or unviable zones stratify the prognosis of the patients as well as PET analysis [56].

Nevertheless, limitations should also be mentioned when dealing with cMRI. The images might be influenced by artefact from movements, while a regular electrocardiographic trace is fundamental in cMRI for a better definition of the morpho-functional characteristics of the cardiac muscle [57].

MRI claustrophobia is a further dilemma when adopting this imaging technique. Although the gadolinium is a safe “contrast” as compared to computer tomography (CT), the claustrophobia may impact on the final adherence of the patient to the imaging technique [58].

## 5. The Therapeutic Options

The rarity of the pSTS-h makes challenging the development of a standardized protocols for a definite treatment. Surgery, chemotherapy, and radiotherapy should be integrated to improve the prognosis.

### 5.1. Surgical Treatment

In the early stage, pSTS-h surgical removal is suitable. After biopsy confirmation for histological characterization, cytoreductive chemotherapy with or without radiotherapy can be optioned. It has been estimated that radical surgery can promote a 51% effective reduction in the overall risk for mortality [59], while cumulative survival rate equal to 72% at 5 years and 59% at 15 years follow-up were in patients treated for cardiac neoplasms, irrespective of the histotypes [60]. No standard techniques can be described due to the rarity of the different histotypes and sites of origin. Orthotopic heart autotransplantation (OHA) has been considered as a possible surgical option [61]—it consists of cardiac explantation, ex vivo tumor resection, reconstruction, and reimplantation [62]. Reardon et al. firstly performed OHA in a large sarcoma of the left atrium and collected a case series on 11 patients [61,63]. Indeed, the literature offers little further data about the application and the outcomes related to this surgical technique [64,65]. Kim et al. [58] evaluated the influence of the site of origin on surgery outcome. They observed an overall 5-year survival rate equal to 17% in patients with pSTS-h involving right cardiac chambers, which was extremely lower as compared to the survival rate from those of the left cardiac chambers [66]. Ramlawi et al. [67] reported a higher prevalence in death between 1 and 6 months after surgery for right cardiac chamber pSTS-h while demonstrating increased prevalence in deaths within 1 month or between 6 and 12 months after surgical removal of left cardiac chamber pSTS-h.

Nevertheless, surgical margins remain the mainstay in understanding the outcome related to surgery—free-cancer cell margins increased the 5-year survival rate of patients with right sarcoma as compared to R1 and/or R2 borders [66]. R1 and/or R2 resections deserve more aggressive interventions and multimodality approaches with chemo- and radiotherapy [68], and heart transplantation is to be considered in selected cases [69]. Moreover, the histotype and the differentiation degree influence the survival—less differentiated neoplasia entails a higher mortality rate and/or recurrence of the disease [70]. Therefore, non-metastatic and localized pSTS-h should undergo complete surgical removal to prolong life expectancy.

### 5.2. Radiotherapy

The radiotherapy (RT) in patients with pSTS-h is questionable due to the risks related to its application. Damages to cardiac muscular cells might be extremely deleterious, thus forcing physicians to adopt reduced or fractioned doses [11], but this may limit the efficacy of the treatment [71]. In the literature, a few case reports are described. Fatima et al. [72] observed a better survival rate in patients who underwent postoperative radiotherapy at 40 to 50 Gy than surgery alone. A retrospective analysis on 168 primary cardiac angiosarcomas selected from the National Cancer Institute’s SEER database confirmed data about outcomes associated with RT. The mean survival rate was higher in patients who underwent radiotherapy (63% higher than controls at univariate analysis) [73].

In the preoperative setting, Thariat et al. considered 124 patients from the French Sarcoma Group, who received 45 Gy in 1.8 Gy per fraction and further implemented with 14 Gy in seven fractions within the residual margins of the lesion, with a result of 15-month mean survival, although six patients developed metastases and/or loco-regional recurrence [71]. Similar results derived from the retrospective analysis by Isambert et al. who observed a 67% improvement in overall survival rate when radiotherapy was considered as a therapeutic option in pSTS-h [8]. Nevertheless, other studies and Aboud et al. did not find an effective role of radiotherapy in ameliorating the outcomes of their patients [29,74].

The application of positron emission tomography/magnetic resonance imaging (PET/MRI)-guided adaptive radiotherapy might become an effective technique able to improve the delivery of the radiation directly to the outer mass, thus trying to avoid damage to healthy cardiac tissue [75]. New techniques, such as intensity-modulated radiation therapy (IMRT) and volumetric modulated arc therapy (VMAT), enable the increase of the radiation dose to 60–70 Gy and minimize the damage to neighbor organs and large vessels, thus achieving greater local regional response [76]. Hong et al. reported administration of intraoperative radiotherapy (IORT: 5 and 15 Gy for localized/locally advanced and metastatic patients, respectively) and external-beam radiotherapy (EBRT: 59.4 and 31.2 Gy for localized/locally advanced and metastatic patients, respectively) in their small, retrospective cohort of 18 cardiac angiosarcomas, although no evaluation about the specific outcome related to the technique was performed due to the managed multimodality approach with chemotherapy and surgery [77].

### 5.3. Chemotherapy

The highly malignant nature of pSTS-h requires evaluation of adjuvant chemotherapy with or without radiotherapy to reduce the cancer mass to plan a definite surgical approach, and, secondly, to prevent the occurrence of metastases [78,79]. Dedicated protocols for chemotherapy in pSTS-h are still a matter of debate due to the paucity of data in the literature and the rarity of these types of cancers. The first-line therapy for pSTS-h is based on Adriamycin at a dose of 75 mg/m^2^ administered as a continuous intravenous infusion over 3 days, and ifosfamide at a dose of 10 g/m^2^ divided over 4 to 5 days [80,81,82]. Adequate hydration therapy should be managed to prevent kidney toxicity, while neurotoxicity related to ifosfamide might be prevented by promoting alkalinization and albumin infusion [82].

The second-line regimen considers a combination of gemcitabine plus docetaxel [81,82]. Generally, gemcitabine might be given on days 1 and 8 at 900 mg/mq, and docetaxel at 100 mg/mq, although a reduced dose may be considered to reduce side effects [82]. Fluid retention due to docetaxel might effectively influence cardiovascular hemodynamics, while gemcitabine should be slowly administrated to allow its phosphorylation for DNA inclusion [82]. Although further chemotherapeutic agents might theoretically be applied in pSTS-h, no randomized controlled trial/studies have been performed, and thus no definite indications may be provided. Frezza et al. [83] retrospectively evaluated the efficacy and safety of chemotherapies (i.e., anthracycline, gemcitabine, pazopanib) in 72 patients with intimal sarcoma. The anthracycline group did not present any cardiac toxicity event, with a real-world overall response rate (rwORR) equal to 38% [83]. Gemcitabine was mostly adopted as a second-line treatment (77% of cases), demonstrating rwORR equal to 8% [83]. The same results (rwORR = 8%) occurred for pazopanib—a tyrosine kinase inhibitor (TKI) with anti-angiogenetic properties—which was considered as a further line therapy in advanced diseases [83].

Temozolomide, dacarbazine, vinorelbine, and liposomal doxorubicin are no longer the first choice in the treatment of advanced or metastatic disease because of their high toxicity profile [84]. Indeed, trabectedin might be adopted as second-line therapy in liposarcoma (LPS) and leiomyosarcoma (LMS) after weighting the risk for sever bone marrow depression and gastrointestinal symptoms.

A phase-III, multicenter, clinical trial compared trabectedin (1.5 mg/mq 24-h i.v.) to dacarbazine (1 g/mq 20–120-min i.v.) in metastatic patients with LMS o LPS after progression to anthracyclines. Patients on trabectedin demonstrated amelioration in PFS (4.2 versus 1.5 months) and 13% reduction in median OS [85]. Similar results were observed from Schöffski et al. [86] who considered eribulin (1.4 mg/mq i.v. on days 1 and 8) on top of anthracycline therapy in patients with LPS or LMS.

Comprehensive management of patients with pSTS-h might consider adequate chemotherapy before surgical excision—in the ESPERO trial (Safety and Efficacy of a Novel Radical Tumor Resection Procedure used in Conjunction with Neoadjuvant Chemotherapy to treat Malignant Primary Right Heart Cardiac Tumors), 33% (8 out of 24) of patients treated with doxorubicin–ifosfamide for 4–6 cycles achieved negative margins with 80% survival at 40 months. Abu Saleh et al. described an improved overall survival and high degree in R0 resections in patients who underwent neoadjuvant chemotherapy before surgery (32 of 44 patients in the study), with an estimated median survival of 20 months compared to 9 months in those without pre-operative systemic therapy [80]. Moreover, upfront surgery followed by adjuvant therapy may provide favorable outcomes in patients with pSTS-h [80,87]. Llombart-Cussac et al. [88] pointed out an overall two-year survival rate equal to 26% in pSTS-h treated with both surgery and post-operative doxorubicin, especially in cardiac angiosarcoma. Hendriksen et al. [89] observed a positive effect of chemotherapy on postoperative 2-year survival after surgical intervention.

Nevertheless, the impact of chemotherapies on the natural history of cardiac sarcomas is not excellent, and hence there is the importance of a multidisciplinary evaluation to define the best therapeutic approach for these patients on the basis of the experience of the group and the patient’s clinic.

The polydrug approach seems to be the most reliable therapy for this type of cancer, above all in the case of sarcomas such as osteosarcoma or rhabdomyosarcoma. Indeed, dedicated trials and further pharmacological advances are needed in order to improve the therapeutic approach to cardiac sarcomas.

## 6. New Therapeutic Strategies and Future Perspectives

The rarity of pSTS-h makes the identification of novel strategies more challenging, but the poor prognosis and delay in diagnosis require a major boost to discover new pharmacological therapies. Zhrebker et al. [90] attempted the whole-exome sequencing of a primary cardiac angiosarcoma. They found the presence of alterations in the gene sequence of *KDR*, i.e., the gene for which transcripts for kinase insert domain receptor (KDR, a type IV receptor tyrosine kinase), also known as vascular endothelial growth factor receptor 2 (VEGFR-2) [90,91]. This led to the suggestion to counteract the activation of vascular endothelial growth factor receptors (bevacizumab and similar). Once more, focal amplification in the *MDM4* gene could be detected in their case, thus leading to the possible application of novel drugs able to counteract p53 binding protein MDM2 [90]. The TAPPAS trial aims to compare pazopanib alone vs. pazopanib and TRC105 (a monoclonal antibody directed towards endoglin, i.e., an angiogenic factor expressed on tumor vessels)—TRC105 increased the antiangiogenic activity of pazopanib, thus potentially reducing the development of the outer mass [92].

Preliminary studies explored the use of the gene transfer model and revealed the possibility to deliver genes—encapsulated into adenoviral vectors—to specific myocardial regions employing a catheter-based percutaneous transluminal approach [93]. Coronary veins may be the perfect road for the catheter to deliver the drug. Further studies will provide more insights. Interesting data include the use of four cardiac hormones, namely, atrial natriuretic peptide, vessel dilator, kaliuretic peptide, and long-acting natriuretic peptide as agents able to counteract tumor growth in vitro [94]. Such molecules can effectively promote the interruption of biochemical cascades, which are usually involved in the mitogenic properties of cancers [94]. As they are usually produced by cardiac structures, the need for evaluating their role as agents able to block the advance of cardiac cancer should be also considered [94].

Immunotherapy is a promising therapeutic alternative [95]. Despite the lack of studies dedicated to cardiac sarcomas, the use of immunotherapies alone or in combination in different type of sarcomas is ongoing. A case series from Florou et al. [96], which involved seven patients with locally advanced or metastatic angiosarcoma treated with checkpoint inhibitors, demonstrated a 71% partial response to drugs. Pembrolizumab with cyclophosphamide in patients with advanced osteosarcomas did not provide significant impact on the overall survival [97]. Apatinib plus camrelizumab (anti-PD1 therapy, SHR-1210) promoted a 50.9% improvement in PFS in patients with osteosarcoma [98]. Combination therapies with immuno-checkpoint inhibitors and oncolytic virus seem to induce anti-tumor immune responses, depending on the sarcoma subtypes [99]. Further studies are needed to define the best therapeutic strategy with these drugs in the setting of pSTS-h.

The cyberknife might be a good option for the surgical removal of pSTS-h. Bonomo et al. [100] used this technique in patients with cardiac angiosarcomas, thus obtaining a reduction in local disease progression. The precision of the cyberknife could be linked to novel 3D heart models [101] that can display on a 3D plane the lesion, thus characterizing the borders, defining the extension of margins, and finally allowing a full resection of it through the cyberknife.

MRI-guided stereotactic body radiotherapy (SBRT) might be considered for palliative purposes and symptom relief in metastatic disease or to eradicate single cardiac metastases or small cardiac masses. MRI guidance limits exposure to radiation and cardiotoxicity, using a dose of 40 Gy in five fractions [102].

Therefore, a multidisciplinary approach to pSTS-h should be managed by considering surgical removal, standard radio/chemotherapies, and target therapies based on genome evaluation [79,103]. Specifically, the “sarcoma team” should be formed by a cardiologist with expertise in cardiac oncology and a multimodality imaging approach to heart visualization and identification of cardiac pathologies. The cardiologist should work in parallel with the cardio-thoracic surgeon and radiologist in order to define the possibility for surgical removal of the outer mass or the need for palliative approaches. Finally, the oncologist will coordinate the final approach—from chemo- to radiotherapies, to target therapies, the oncologist should identify the correct approach to the mass and the type of therapy to be addressed to eradicate the tumor and prevent recurrences.

Indeed, the rarity of the disease, the complexity of the treatments, and the need for a comprehensive surgical and pharmacological management of pSTS-h induce the need to address patients to dedicated medical centers. Therefore, the identification of the most skillful hospital in pSTS-h treatment should be the basis for the final diagnosis and treatment of pSTS-h, avoiding delays and soft approaches to this disease.

## 7. Conclusions

Primary cardiac sarcomas of the heart are rare neoplasms with a negative prognosis due to their aggressive nature. Surgery remains the gold standard technique for pSTS-h, above all when succeeding in removing cancer cells even from the margins of the lesion. Indeed, despite innovations in surgery and chemo-/radiotherapies, the survival rate of the patients is challenging. The multimodality approaches combining complete resection of cancer and the adoption of chemo-/radiotherapies is the best approach to increase overall survival and progression-free survival.

## Figures and Tables

**Figure 1 biomedicines-09-00774-f001:**
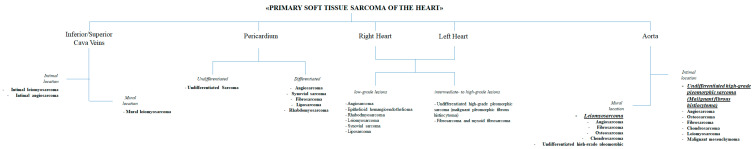
Classification of the primary soft tissue sarcoma of the heart. The figure gathers the combination of the surgical and histologic approach to the classification of these neoplasms. Cardiac sarcomas are differentiated according to the site of origin into cardiac chambers and great arteries/veins; then, sarcomas are identified in agreement with their own histologic nature.

**Figure 2 biomedicines-09-00774-f002:**
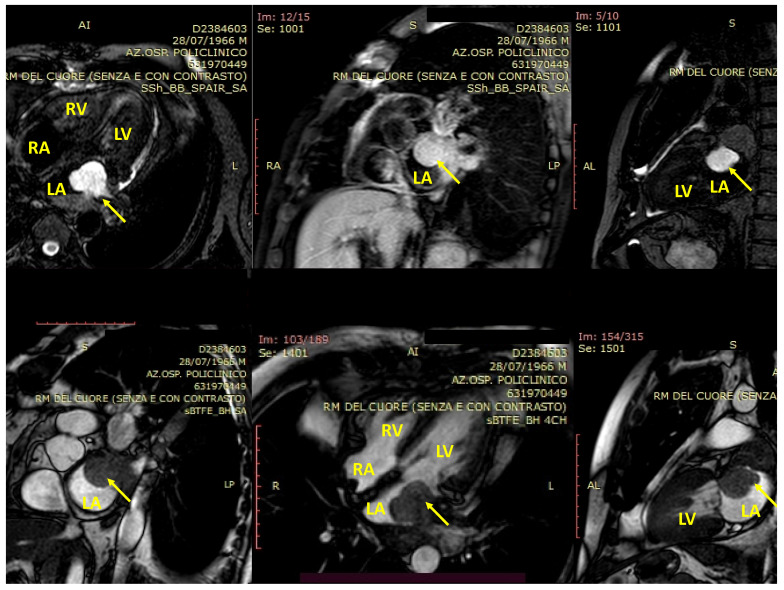
Cardiac magnetic resonance imaging of a cardiac sarcoma of the left atrium (arrow). The images revealed the presence of a cardiac sarcoma of the left atrium, which encroached upon the cardiac chamber with its sessile branch (dimensions: 4 × 4 cm). The origin of the mass rose up from the posterior wall of the atrium and extended into the left pulmonary veins, with no detectable cleavage planes to aortic and esophageal walls. The lesion demonstrated late and not homogeneous enhancement. LA: left atrium; LV: left ventricle; RA: right atrium; RV: right ventricle.

**Figure 3 biomedicines-09-00774-f003:**
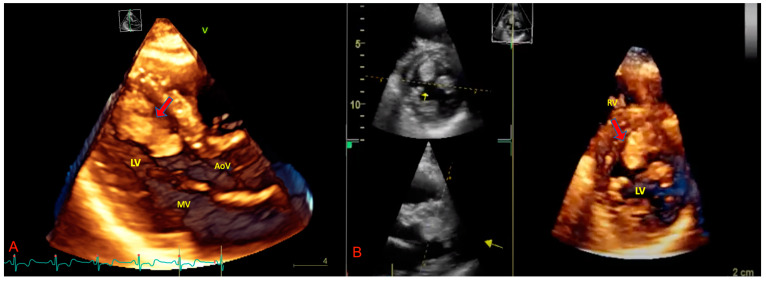
Three-dimensional transthoracic echocardiography in apical three-chamber view (**A**) and parasternal short-axis view (**B**) showing intracardiac sarcoma attached to left ventricular apex. The arrows (red and yellow) indicate the cancer mass. AoV: aortic valve; LV: left ventricle; MV: mitral valve; RV: right ventricle.

**Figure 4 biomedicines-09-00774-f004:**
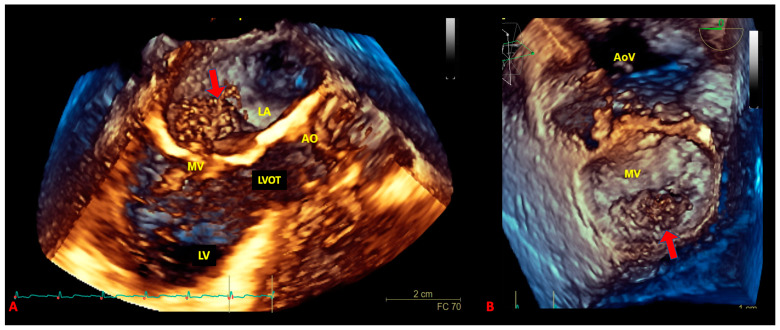
Three-dimensional transoesophageal echocardiography in long-axis view (**A**) and en-face view (**B**) showing intracardiac sarcoma involving the atrial side of the posterior leaflet (P1 and P2 scallops) of the mitral valve. The arrows (red) indicate the cancer mass.AO: aorta; AoV: aortic valve; LA: left atrium; LV: left ventricle; LVOT: left ventricular outflow tract; MV: mitral valve.

**Table 1 biomedicines-09-00774-t001:** Mean incidence in primary cardiac sarcomas among all cardiac tumors according to international registries. Data were from the Armed Forces Institute of Pathology (AIFP), the Japanese Circulation Society (JCS), and the Japanese Association for Thoracic Surgery (JATS) registries [3,17,18,19]. The French Sarcoma Group registry reported the percentage of patients each kind of sarcoma among all of the primary cardiac sarcomas (PCSs) collected by the group [8].

Type of Sarcoma	AFIP Registry Period: <1974–1993	JCS Registry Period: 1999–2010	JATS Registry Period: 1999–2010	French Sarcoma Group Period: 1977–2010 % among PCSs
Angiosarcoma	7.3–8.5%	9.5%	8.2%	32.3%
Undifferentiated high grade pleomorphic sarcoma	4.1–8.5%%	3.3%	4.3%	36.3%
Leiomyosarcoma	0.19–3.1%	1.9%	2.2%	12.9%
Rhabdomyosarcoma	1.6–4.9%	0.8%	0.45%	18.6%
Synovial sarcoma	0.19–1.4%	0.8%	0.9%	
Liposarcoma	0.19–0.5%	1.1%	0.9%	
Fibrosarcoma and myxoid fibrosarcoma	2.3–3.2%	0.3%	0%	
Osteosarcoma/Chondrosarcoma	0.94–4.7%	0–0.8%	0.45–1.3%	

## Data Availability

Not applicable.

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
