# Peer review of "Primary Soft Tissue Sarcoma of the Heart: An Emerging Chapter in Cardio-Oncology"

_biomedicines, 2021, doi:10.3390/biomedicines9070774_

Round 1

Reviewer 1 Report

The current review article on the clinical presentation, diagnosis, treatment and prognosis of cardiac sarcomas is very well written and provides a good overview of the field. I would welcome representative CT images of cardiac sarcomas. I suggest that the authors rephrase the second sentence in the introduction, since referral to myxomas, which are benign tumours, is misleading. 

Author Response

  1. The current review article on the clinical presentation, diagnosis, treatment and prognosis of cardiac sarcomas is very well written and provides a good overview of the field. I would welcome representative CT images of cardiac sarcomas. I suggest that the authors rephrase the second sentence in the introduction, since referral to myxomas, which are benign tumours, is misleading.

We would like to really thank the reviewer for his/her comments and appreciation of this review. Unfortunately, we did not have any CT images of cardiac sarcomas. All the images are from cases admitted to our Departments but we have no CT data. We did not include images from the web due to copyright laws. Finally, thanks for the suggestion to remove “myxomas” from the introduction section: we rephrase the sentence.

Reviewer 2 Report

Dear Authors, 

Nice review of a very rare condition where the experience is very limited and summaries like the present, may be of great help for physicians. 

Somme comments:

  1. Please update  bibliography. For ex WHO soft tissue & Bone sarcoma is in its 5th edition ( 2020).
  2. I found very useful the description of the different image technics available and pros/cons but I miss MRI ( you provide an image but not data on the technic itself) or PET-CT. Since MRI is one of the most important tools we have for primary sarcomas in general, it deserves an specific place. 
  3. Please label properly fig 2 ( like you did for fig 3 and 4: initials and arrows). 
  4. Worth mentioning the pattern of dissemination in comparison with sarcoma of other primary sites.
  5. Regarding chemo, if you include docetaxel-gem and pazo as 2nd line, you should also mention DTIC-gemcitabine, Trabectedin or Eribulin, as we have the same minimun level of evidence for each one. As for sarcoma in other localizations, no direct comparison has been done. We use all of them depending on several factors like ECOG and histology. Also mention that rhabdo ( embryonal or alveolar) receive a totally different chemo approach (polydrug) than other soft part sarcomas.  Osteo usually needs a different approach also.  
  6. For future directions, it could be useful to mention SBRT and regarding systemic treatment also mention inmunotherapy ( for ex s sarcoma & myxoid liposarcoma could benefit from TCR, angiosarcoma maybe from checkpoint inh, etc). 
  7. As an important part of the authors belong to other specialities different from oncology you could exploit this multidisciplinary team: need for follow up from the carioncology point of view, options of reconstructions, criteria for declaring the tumor irresecable, etc
  8. Finally, you should stress that this patients should only be "touched" ( included diagnostic and treatment) at sarcoma expert centers. There´s extensive literature supporting this for most sarcoma, but it´s specialy relevant for rare localizations. 

Author Response

Reviewer #2

We thank this Reviewer for her/his useful suggestions. We sincerely appreciate his/her comments on our work. This is our point-to-point reply:

  1. Please update bibliography. For ex WHO soft tissue & Bone sarcoma is in its 5th edition ( 2020).

Thank you for the suggestion. We updated some references and the WHO one as indicated.

  1. I found very useful the description of the different image technics available and pros/cons but I miss MRI ( you provide an image but not data on the technic itself) or PET-CT. Since MRI is one of the most important tools we have for primary sarcomas in general, it deserves an specific place.

Thank you very much for this comment. We included a dedicated paragraph to cardiac MRI in the setting of pSTS-h.

  1. Please label properly fig 2 ( like you did for fig 3 and 4: initials and arrows).

Thank you for this comment. We updated figure 2.

  1. Worth mentioning the pattern of dissemination in comparison with sarcoma of other primary sites.

Thank you very much for the suggestion. We included a few sentences in order to better outline the metastatic behaviour of the sarcomas of the heart as compared to those arising from other tissues.

  1. Regarding chemo, if you include docetaxel-gem and pazo as 2nd line, you should also mention DTIC-gemcitabine, Trabectedin or Eribulin, as we have the same minimun level of evidence for each one. As for sarcoma in other localizations, no direct comparison has been done. We use all of them depending on several factors like ECOG and histology. Also mention that rhabdo ( embryonal or alveolar) receive a totally different chemo approach (polydrug) than other soft part sarcomas.  Osteo usually needs a different approach also.

Thank you once again for this suggestions. Effectively, little data are about the role of chemotherapies in cardiac sarcomas and the widest experience of clinicians are based on the treatment of general sarcomas. We updated the section in order to improve the comprehensive management of cardiac sarcomas.

  1. For future directions, it could be useful to mention SBRT and regarding systemic treatment also mention inmunotherapy ( for ex s sarcoma & myxoid liposarcoma could benefit from TCR, angiosarcoma maybe from checkpoint inh, etc).

We really thank the reviewer for his/her comments. We improve the section in order to better assess the role of immunotherapies in the setting of cardiac sarcomas.

  1. As an important part of the authors belong to other specialities different from oncology you could exploit this multidisciplinary team: need for follow up from the carioncology point of view, options of reconstructions, criteria for declaring the tumor irresecable, etc.

Thank you very much for this suggestion. We updated the section dedicated to the multidisciplinary team in order to better understand the role of different specialists in the setting of the treatment of cardiac sarcomas.

  1. Finally, you should stress that this patients should only be "touched" ( included diagnostic and treatment) at sarcoma expert centers. There´s extensive literature supporting this for most sarcoma, but it´s specialy relevant for rare localizations.

That’s truly right. The need for the identification of dedicated centres which have the experience to manage pSTS-h is fundamental for the trying to promptly diagnose and treat these diseases. We included a dedicated sentence in the paper in order to stress such a point.